# The successful and safe conversion of joint arthroplasty to same-day surgery: A necessity after the COVID-19 pandemic

Steven Habbous[1,2]*, James Waddell[3], Erik Hellsten[1]

**1** Ontario Health (Strategic Analytics), Toronto, Ontario, Canada, **2** Epidemiology & Biostatistics, Western University, London, Ontario, Canada, **3** Division of Orthopedic Surgery, St. Michael's Hospital, Toronto, Ontario, Canada

* shabbous@uwo.ca

## Abstract

### Introduction

A key strategy to address system pressures on hip and knee arthroplasty through the COVID-19 pandemic has been to shift procedures to the outpatient setting.

### Methods

This was a retrospective cohort and case-control study. Using the Discharge Abstract Database and the National Ambulatory Care Reporting System databases, we estimated the use of outpatient hip and knee arthroplasty in Ontario, Canada. After propensity-score matching, we estimated rates of 90-day readmission, 90-day emergency department (ED) visit, 1-year mortality, and 1-year infection or revision.

### Results

204,066 elective hip and 341,678 elective knee arthroplasties were performed from 2010–2022. Annual volumes of hip and knee arthroplasties increased steadily until 2020. Following the start of the COVID-19 pandemic (March 1, 2020) through December 31, 2022 there were 7,561 (95% CI 5,435 to 9,688) fewer hip and 20,777 (95% CI 17,382 to 24,172) fewer knee replacements performed than expected. Outpatient arthroplasties increased as a share of all surgeries from 1% pre-pandemic to 39% (hip) and 36% (knee) by 2022. Among inpatient arthroplasties, the tendency to discharge to home did not change since the start of the pandemic. During the COVID-19 era, patients receiving arthroplasty in the outpatient setting had a similar or lower risk of readmission than matched patients receiving inpatient arthroplasty [hip: RR 0.65 (0.56–0.76); knee: RR 0.86 (0.76–0.97)]; ED visits [hip: RR 0.78 (0.73–0.83); knee: RR 0.92 (0.88–0.96)]; and mortality, infection, or revision [hip: RR 0.65 (0.45–0.93); knee: 0.90 (0.64–1.26)].

### Conclusion

Following the start of the COVID-19 pandemic in Ontario, the volume of outpatient hip and knee arthroplasties performed increased despite a reduction in overall arthroplasty volumes.

**Data Availability Statement:** Data availability: Ontario Health is prohibited from making the data used in this research publicly accessible if it includes potentially identifiable personal health

information and/or personal information as defined in Ontario law, specifically the Personal Health Information Protection Act (PHIPA) and the Freedom of Information and Protection of Privacy Act (FIPPA). Due to these legal and ethical restrictions, data will not be made publicly available. However, upon request, data de-identified to a level suitable for public release may be provided. Requests can be made to OH-CCO_Research@ontariohealth.ca.

**Funding:** The authors received no specific funding for this work.

**Competing interests:** The authors have declared that no competing interests exist.

This shift in surgical volumes from the inpatient to outpatient setting coincided with pressures on hospitals to retain inpatient bed capacity. Patients receiving arthroplasty in the outpatient setting had relatively similar outcomes to those receiving inpatient surgery after matching on known sociodemographic and clinical characteristics.

## Introduction

Arthroplasties are among the most frequently performed surgeries, owing to their safety and effectiveness for relieving major joint pain and stiffness primarily caused by advanced osteoarthritis. At the onset of the COVID-19 pandemic in Ontario, elective surgeries were temporarily suspended [1]. As a result, many patients who otherwise might have received a hip or knee arthroplasty had their surgeries delayed or deferred, with potential detrimental impacts on patient quality-of-life and risk of complications [2–4].

With continued strains on hospital bed capacity and reduced health system resources since the start of the COVID-19 pandemic, efforts to reconcile the surgical backlog may require a shift in patient management from the inpatient to outpatient setting. Hip and knee arthroplasties are excellent candidate surgeries to reduce the burden on hospital inpatient wards as studies have shown many of these procedures can be safely performed in an outpatient setting without compromising patient outcomes [5–7]. Recognizing this, the Centers for Medicare and Medicaid Services in the United States removed from the inpatient-only list total knee replacement in 2018 and total hip replacement in 2020, resulting in a drastic shift towards outpatient arthroplasty [8, 9]. Emergency department visits and readmissions were favourable following outpatient hip but worse following outpatient knee replacement. A systematic review of the literature comparing inpatient with outpatient hip and knee arthroplasty captured studies predominantly from the United States, and studies from other countries were sparse (two from Canada and three from Europe) and not population-based (study size <600).

In the present study, we examine how the provision of outpatient hip and knee arthroplasty in Ontario has changed between 2010 and 2022, with a focus on the COVID-19 pandemic period. This is the largest population-based analyses to date from a universal health care system on this topic. These findings can be used to inform strategies to plan for surgical recovery.

## Methods

In this retrospective population-based cohort study, inpatient and outpatient hip and knee arthroplasty surgeries were captured between 2010 and 2022. We examined 1) the effect of the pandemic on surgical volumes; 2) trends of inpatient versus outpatient arthroplasty; and 3) outcomes comparing inpatient with outpatient arthroplasty.

### Cohort definitions

Inpatient and outpatient arthroplasties performed in Ontario hospitals between January 1, 2010 (the earliest data available for comorbidity assessment) and December 31, 2022 were identified from the Discharge Abstract Database (inpatient) and National Ambulatory Care Reporting System (outpatient) databases using the Canadian Classification of Interventions (CCI) codes for a main intervention of hip arthroplasty (1.VA.53 for hip joint; 1.SQ.53 for pelvis) or knee arthroplasty (1.VG.53 for knee joint; 1.VP.53 for patella) (S1 Table in S1 File). Data were extracted on June 20, 2023. Urgent procedures were removed, defined as either an

admission record with entry code = "E", admit category = "U", ambulance arrival, or if the outpatient record was flagged as an emergency (derived from MIS functional codes).

Arthroplasties identified from the Discharge Abstract Database (inpatient) but having a discharge date equal to the arthroplasty date were reclassified as outpatient procedures, as these may have been either miscoded as inpatient or initially planned for inpatient. This decision was made *a priori*, supported by the data, and cautioned by the NACRS data quality documentation (S1 Fig in S1 File) [10].

All hospitals in Ontario are mandated to report all admission records (discharge abstracts) to the CIHI-DAD. Coding is performed by trained nosologists and routine data quality checks are performed [11]. All surgical outpatient hospital visits are reported to either the CIHI-DAD or CIHI-NACRS as a Level 3 submission, which mandates ICD-10-CA and CCI coding [10].

## Covariates

Procedures were classified as total or partial arthroplasties using the CCI code reporting the type of implant as either dual- or tri-component device (total arthroplasty) or single-component device (partial arthroplasty) [12].

The diagnosis related to the joint replacement was assigned using the most responsible diagnosis for the repair. For 5,574 (0.9%) procedures that had no diagnostic code, we imputed the diagnosis as rheumatoid arthritis (RA) if the patient had any hospital encounter for RA from DAD/NACRS in the previous 3 years or at least 3 physician billings over the previous 2 years with at least one being assigned by a specialist (sensitivity = 78%) [13]. The remainder were imputed as osteoarthritis if there was any physician billing or hospital record within the prior 3 years (sensitivity = 77%) [14].

Comorbidity was estimated using the Charlson Comorbidity Index. This was modified to 1) include oral agents for diabetes from the Ontario Drug Benefits database (for Ontarians age 65+ years) [15]; and 2) exclude rheumatoid arthritis from the connective tissue or rheumatoid disease component of the Charlson score, since this was treated as a covariate (S2 and S3 Tables in S1 File).

Neighbourhood sociodemographics were derived from the 2016 Census and the Ontario Marginalization Index, which included rurality, material deprivation (composite of education, lone-parent families, income, employment, housing in disrepair) and ethnic diversity (composite of immigration status and visible minority) [16]. For admitted patients, discharge home was defined as discharge disposition '04' or '05'.

## Outcomes during the COVID-19 era by setting

To assess whether short-term outcomes were different for inpatient compared with outpatient arthroplasty patients, a sub-cohort was created, restricted to unilateral elective (non-urgent) primary repairs performed since March 1, 2020 (approximate start of the COVID-19 pandemic) for osteoarthritis or rheumatoid arthritis. Procedures initially classified as a primary based on the status attribute code were excluded if there was evidence of a prior ipsilateral arthroplasty since January 1, 2007 due to more accurate coding of laterality [12]. We also excluded primary arthroplasties with hardware removal within 30 days prior (CCI code 1VG55, 1VP55, 1SQ55, 1VA55 in any position).

Outcomes included a composite of joint infection, revision arthroplasty for any reason, or all-cause mortality within 1 year of the procedure (restricted to patients having an arthroplasty between March 1, 2020 and December 31, 2021 for sufficient follow-up). Joint infection was defined using T845.3 (hip) or T845.4 (knee) in any position from DAD/NACRS [17]. All-cause mortality was obtained from the Registered Persons Database. Secondary outcomes included

90-day readmissions and 90-day unplanned emergency department visits. Readmissions and ED visits were defined as any admission or ED visit within 90 days of discharge (inpatient setting) or registration date (outpatient setting). Unplanned ED visits included those where the MIS functional centre code started with 7*310 and had an ED visit indicator flag.

## Statistical methods

Descriptive statistics were used to estimate changes over time. To calculate the difference between the number of arthroplasties performed during the pandemic period had there been no pandemic, we forecasted the pre-pandemic weekly arthroplasty counts using linear regression with the covariates for year (general trends), month (seasonal trends), and week number (holidays and other regular fluxes in surgical activity). The expected number of procedures performed between 2020 and 2022 was extrapolated from the pre-pandemic model and the observed values were subtracted from the expected values.

To describe the differences between inpatient and outpatient procedures, we present standardized differences and odds ratios (OR) with 95% confidence intervals (CI) from logistic regression.

To compare outcome by setting, patients receiving a hip or knee arthroplasty in the outpatient setting were propensity-matched to those performed in the inpatient setting. The propensity score was estimated using logistic regression with age, sex, rurality, deprivation quintile, instability quintile, dependency quintile, ethnic diversity quintile, comorbidity score, total/partial repair, and repair date. Nearest-neighbour matching with a caliper of 0.05 was performed using the matchit() function in the MatchIt package in R (attempting to match 2 inpatient procedures to 1 outpatient procedure). Balance was assessed using standardized mean differences (all $<|0.1|$, demonstrating good balance). For patient outcomes, modified Poisson regression was used to estimate the risk ratio (RR) with 95% CI using robust standard error estimation using the matched dataset [18]. In sensitivity analysis, the association between setting and patient outcomes were compared using different statistical models, including crude, adjusted, crude after matching, and adjusted after matching (adjusted for all variables used to generate the propensity score). Analysis was performed on complete-case.

## Privacy and software

All analyses were performed using SAS v9.4 (SAS Institute Inc., Cary, NC) or RStudio (1.2.5). Values <6 were suppressed to prevent re-identification. This study was compliant with section 45(1) of PHIPA (Ontario Health is a prescribed entity): ethics review was not required. Data were analyzed using the Analytics Data Hub at Ontario Health with patient identifiers removed or pseudonymized prior to access.

## Results

After exclusions (Fig 1), there were 204,066 elective hip and 341,678 elective knee arthroplasties performed in Ontario since 2010, most of which were primary repairs (70% hip; 67% knee), total replacements (97% hip; 98% knee), and performed on women (55% hip; 61% knee) (S4 Table in S1 File) The most common responsible diagnosis was osteoarthritis (90% hip; 94% knee).

## Volumes

Pre-pandemic (2010–2019), the year-over-year increase in the number of arthroplasties were 748 (SE 24.9) for hip and 1,151 (SE 54.3) for knee (Fig 2). After accounting for year-over-year

Identify all procedures from DAD and NACRS with hip (1SQ53, 1VA53) or knee (1VG53, 1VP53) arthroplasty (N=627,417 procedures; n=464,460 unique patients) between January 1, 2010 and December 31, 2022

- Exclude
    - 4077 (0.7%) primary procedure is insertion of cement spacer
    - 368 (<1%) unknown procedure type (not classified as primary or revision). For procedures before FY2012, only revisions were coded. All missing procedure types were assumed to be primary and were retained.
    - 116 (<1%) patients aged <18 or >105 years
    - 110 (<1%) patients with duplicate conflicting records on the same day
    - 53 (<1%) missing laterality

N=622,471 procedures (n=463,597 unique patients)

- Exclude urgent repairs
    - Entry code = E
    - Admit category = U
    - Arrived by ambulance

N=545,744 procedures (n=403,359 unique patients)

- For analysis in COVID-19 era, include
    - Joint replacements occurring between March 1, 2020 and December 30, 2022
    - Most responsible diagnosis osteoarthritis or rheumatoid arthritis
    - Primary surgery (attribute status = "P" or "0", no prior ipsilateral arthroplasty since 2007, and no evidence of device removal within 30 days prior to arthroplasty)
    - Known unilateral as left or right
    - Keep the first procedure by joint and laterality

N=96,410 procedures (n=96,410 unique patients)

**Fig 1. Patient selection.**

changes, monthly seasonality, and weekly trends, assuming a continued trajectory through March 2020 to December, 2022, there were 7,561 (95% CI 5,435 to 9,688) fewer elective hip and 20,777 (95% CI 17,382 to 24,172) fewer elective knee replacements performed than expected (S2 Fig in S1 File).

## Outpatient versus inpatient

Outpatient arthroplasties increased from 1% pre-pandemic to 39% (hip) and 36% (knee) by 2022 (Fig 2). Between March 2020 and December 2022, patients were more likely to receive a hip or knee arthroplasty in the outpatient versus the inpatient setting if they were younger [OR 0.70 (0.69–0.71) for hip; OR 0.74 (0.72–0.75) for knee], male [OR 1.21 (1.16–1.26) for hip; OR 1.21 (1.17–1.25) for knee], received a primary arthroplasty versus a revision [OR 1.48 (1.38–1.58) for hip; OR 1.29 (1.23–1.36) for knee], received a partial versus total arthroplasty [OR 1.64 (1.37–1.96) for hip; OR 2.46 (2.18–2.77) for knee], and had a lower comorbidity score

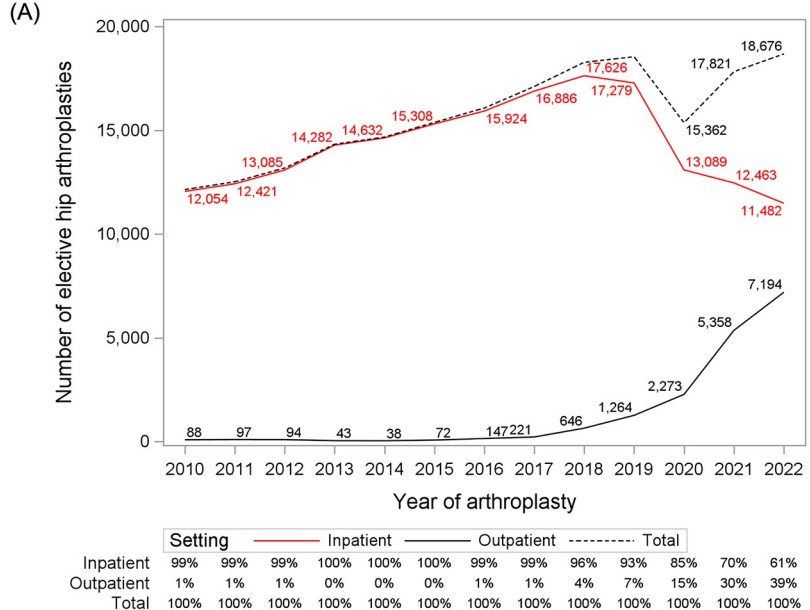

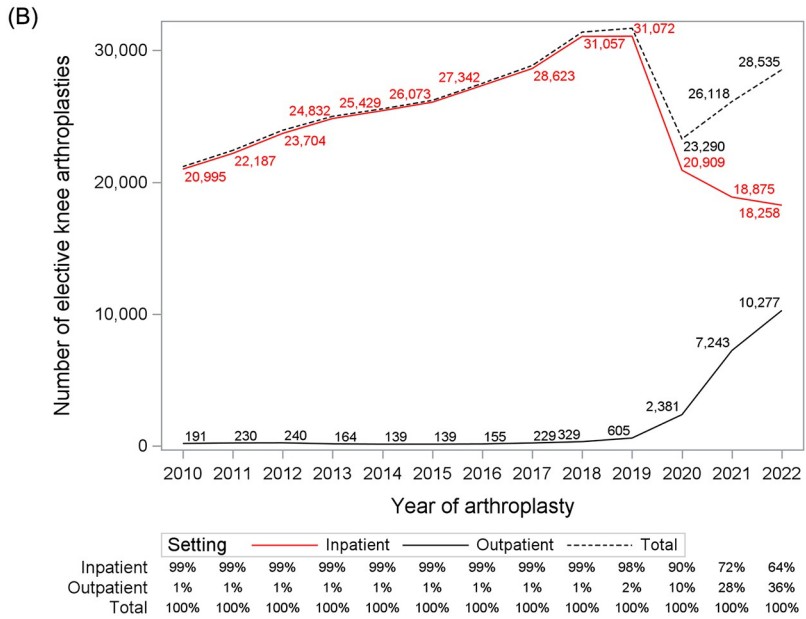

**Fig 2. Elective arthroplasties by setting over time.** Number of elective hip (A) and knee (B) arthroscopies over time and setting.

[OR 0.22 (0.18–0.27) for hip; OR 0.28 (0.24–0.33) for knee for 3+ comorbidities versus none] (Table 1). Some differences by joint were observed for socio-demographic factors. For hip replacements, outpatient procedures were least likely for patients residing in neighbourhoods with the highest deprivation [OR 0.64 (0.59–0.69)], highest dependency [OR 0.87 (0.80–0.94)], and highest ethnic diversity [OR 0.84 (0.77–0.91)]. However, for knee arthroplasty, outpatient procedures were most likely among patients residing in neighbourhoods of the highest ethnic

**Table 1. Characteristics of repairs by joint and setting during the COVID-19 era (March 2020–December 2022).**

| | Hip arthroplasty | | | | Knee arthroplasty | | | |
|---|---|---|---|---|---|---|---|---|
| | Inpatient (n = 33,929) | Outpatient (n = 14,555) | Std diff | OR (95% CI)[a] N = 37,650 | Inpatient (n = 52,620) | Outpatient (n = 19,696) | Std diff | OR (95% CI)[a] N = 55,680 |
| Patient demographics | | | | | | | | |
| Age (years) | 69.4 (SD 10.9) | 64.9 (SD 10.1) | -0.42 | 0.70 (0.69–0.71) | 69.3 (SD 8.8) | 66.8 (8.3) | -0.29 | 0.74 (0.72–0.75) |
| Male vs female | 14,872 (44%) | 7,344 (50%) | 0.13 | 1.21 (1.16–1.26) | 20,795 (40%) | 8,464 (43%) | 0.07 | 1.21 (1.17–1.25) |
| Procedure characteristics | | | | | | | | |
| Primary (modified) vs revision arthroplasty | 27,762 (82%) | 12,945 (89%) | 0.20 | 1.48 (1.38–1.58) | 42,602 (81%) | 16,770 (85%) | 0.11 | 1.29 (1.23–1.36) |
| Partial vs total arthroplasty | 854 (2.5%) | 219 (1.5%) | -0.07 | 1.64 (1.37–1.96) | 1,031 (2.0%) | 574 (2.9%) | 0.06 | 2.46 (2.18–2.77) |
| Bilateral vs unilateral | 493 (1.5%) | 43 (0.3%) | -0.12 | 0.12 (0.09–0.17) | 1,297 (2.5%) | 107 (0.5%) | -0.16 | 0.17 (0.14–0.20) |
| Clinical characteristics | | | | | | | | |
| Comorbidity | | | | | | | | |
| 0 (none) | 23,781 (70%) | 12,409 (85%) | 0.39 | 1.0 (ref) | 34,958 (66%) | 15,648 (79%) | 0.32 | 1.0 (ref) |
| 1 | 6,819 (20%) | 1,749 (12%) | | 0.57 (0.54–0.61) | 12,387 (24%) | 3,319 (17%) | | 0.63 (0.60–0.65) |
| 2 | 2,164 (6%) | 198 (2%) | | 0.33 (0.29–0.37) | 3,673 (7%) | 553 (3%) | | 0.36 (0.33–0.40) |
| 3+ | 1,165 (3%) | 99 (1%) | | 0.22 (0.18–0.27) | 1,602 (3%) | 176 (1%) | | 0.28 (0.24–0.33) |
| Osteoarthritis as most responsible diagnosis[b] | 30,437 (90%) | 14,035 (96%) | 0.27 | 2.64 (2.36–2.95) | 49,210 (94%) | 19,047 (97%) | 0.15 | 2.11 (1.91–2.34) |
| Patient socio-demographics | | | | | | | | |
| Rurality | | | | | | | | |
| Urban | 28,136 (84%) | 11,909 (82%) | | 1.0 (ref) | 43,768 (84%) | 16,449 (84%) | | 1.0 (ref) |
| Rural | 5,543 (16%) | 2,542 (18%) | 0.03 | 1.08 (1.02–1.15) | 8,471 (916%) | 3,115 (16%) | -0.05 | 1.01 (0.96–1.07) |
| Missing | 250 (0.7%) | 104 (0.7%) | | NR | 381 (0.7%) | 132 (0.7%) | | NR |
| Deprivation | | | | | | | | |
| 1 (least marginalized) | 8,275 (25%) | 4,415 (31%) | 0.18 | 1.0 (ref) | 11,688 (23%) | 4,588 (24%) | 0.07 | 1.0 (ref) |
| 2 | 7,254 (22%) | 3,388 (24%) | | 0.87 (0.82–0.92) | 11,055 (21%) | 4,348 (22%) | | 1.00 (0.95–1.06) |
| 3 | 6,603 (20%) | 2,724 (19%) | | 0.78 (0.73–0.83) | 10,371 (20%) | 4,043 (21%) | | 1.01 (0.96–1.06) |
| 4 | 6,098 (18%) | 2,203 (15%) | | 0.71 (0.66–0.76) | 10,000 (19%) | 3,498 (18%) | | 0.94 (0.88–0.99) |
| 5 (most marginalized) | 5,140 (15%) | 1,582 (11%) | | 0.64 (0.59–0.69) | 8,644 (17%) | 2,935 (15%) | | 0.94 (0.88–1.00) |
| Missing | 559 (1.7%) | 243 (1.7%) | | NR | 862 (1.6%) | 284 (1.4%) | | NR |
| Instability | | | | | | | | |
| 1 (least marginalized) | 5,314 (16%) | 2,745 (195) | 0.17 | 1.0 (ref) | 9,042 (17%) | 4,128 (21%) | 0.15 | 1.0 (ref) |
| 2 | 6,824 (20%) | 3,404 (24%) | | 1.05 (0.98–1.12) | 10,818 (21%) | 4,438 (23%) | | 0.97 (0.92–1.02) |
| 3 | 7,178 (22%) | 3,157 (22%) | | 0.99 (0.92–1.06) | 11,378 (22%) | 4,308 (22%) | | 0.93 (0.88–0.99) |
| 4 | 6,638 (20%) | 2,578 (18%) | | 0.97 (0.90–1.04) | 10,270 (20%) | 3,441 (18%) | | 0.84 (0.79–0.89) |
| 5 (most marginalized) | 7,416 (22%) | 2,428 (17%) | | 0.91 (0.84–0.98) | 10,250 (20%) | 3,097 (16%) | | 0.78 (0.74–0.84) |
| Missing | 559 (1.7%) | 243 (1.7%) | | NR | 862 (1.6%) | 284 (1.4%) | | NR |
| Dependency | | | | | | | | |
| 1 (least marginalized) | 4768 (14%) | 2519 (18%) | 0.16 | 1.0 (ref) | 8325 (16%) | 3,739 (19%) | 0.11 | 1.0 (ref) |
| 2 | 5467 (16%) | 2471 (17%) | | 0.92 (0.86–0.99) | 8,567 (17%) | 3,489 (18%) | | 0.99 (0.93–1.05) |
| 3 | 6032 (18%) | 2852 (20%) | | 1.00 (0.93–1.08) | 9,216 (18%) | 3,625 (19%) | | 0.99 (0.93–1.05) |
| 4 | 6660 (20%) | 2661 (19%) | | 0.85 (0.79–0.92) | 10,094 (20%) | 3,621 (19%) | | 0.93 (0.87–0.99) |
| 5 (most marginalized) | 10443 (31%) | 3809 (27%) | | 0.87 (0.80–0.94) | 15,556 (30%) | 4,938 (25%) | | 0.92 (0.86–0.98) |
| Missing | 559 (1.7%) | 243 (1.7%) | | NR | 862 (1.6%) | 284 (1.4%) | | NR |
| Ethnic diversity | | | | | | | | |
| 1 (least diverse) | 8,994 (27%) | 3,872 (27%) | 0.04 | 1.0 (ref) | 13,967 (27%) | 4,798 (25%) | 0.13 | 1.0 (ref) |
| 2 | 7,777 (23%) | 3,266 (23%) | | 0.91 (0.86–0.97) | 11,295 (22%) | 4,122 (21%) | | 1.04 (0.99–1.10) |
| 3 | 6,859 (21%) | 3,052 (21%) | | 0.91 (0.85–0.97) | 9,606 (19%) | 3,572 (18%) | | 1.05 (0.99–1.11) |

*(Continued)*

**Table 1.** (Continued)

| | Hip arthroplasty | | | | Knee arthroplasty | | | |
|---|---|---|---|---|---|---|---|---|
| | Inpatient (n = 33,929) | Outpatient (n = 14,555) | Std diff | OR (95% CI)[a] N = 37,650 | Inpatient (n = 52,620) | Outpatient (n = 19,696) | Std diff | OR (95% CI)[a] N = 55,680 |
| 4 | 5,820 (17%) | 2,595 (18%) | | 0.90 (0.83–0.97) | 8,634 (17%) | 3,112(16%) | | 1.02 (0.96–1.09) |
| 5 (most diverse) | 3,920 (12%) | 1,526 (11%) | | 0.84 (0.77–0.91) | 8,256 (16%) | 3,808 (20%) | | 1.37 (1.28–1.46) |
| Missing | 559 (1.7%) | 243 (1.7%) | | NR | 862 (1.6%) | 284 (1.4%) | | NR |

Std diff—standardized difference; OR—odds ratio; CI—confidence interval

[a] Adjusted for all variables shown; N is the complete-case count; NR—not reported due to small size (cells <6 were suppressed), collinearity (e.g. most responsible diagnosis and primary/revision), or missing; p-values <0.0001 except for hip (sex p = 0.0001; instability p = 0.02; urban p = 0.76) and knee (deprivation p = 0.01; dependency p = 0.0005; urban p = 0.0005)

[b] missing and other category suppressed due to small count

diversity [OR 1.37 (1.28–1.46)] and least likely for the highest instability [OR 0.78 (0.74–0.84)]. There was significant regional variation during the pandemic era (Fig 3A and 3B).

## Discharge disposition

Since March 2020, most patients were discharged home following an inpatient procedure (93% hip; 96% knee), with some regional variability (Fig 3C and 3D). There was no indication that this increased since the pandemic (Fig 4).

## Outcomes

Outcomes between outpatient and inpatient settings were compared during the COVID-era (March 2020-December 2022), restricted to primary (non-revision) unilateral replacements performed for osteoarthritis or rheumatoid arthritis (n = 38,846 hip; n = 57,564 knee).

**Readmissions.** The 90-day readmission rates were 4.9% and 2.4% following inpatient and outpatient hip arthroplasty, respectively [crude RR 0.48 (0.42–0.55)] (Table 2). For knee arthroplasty, the readmission rates were 4.1% and 2.9% following inpatient and outpatient repair, respectively [crude RR 0.71 (0.63–0.79)]. After matching, patients receiving outpatient hip or knee arthroplasty were less likely to be readmitted within 90 days compared with inpatients [RR 0.65 (0.56–0.76) for hip; RR 0.86 (0.76–0.97) for knee]. Results were similar between different statistical models (S5 Table in S1 File).

**ED visits.** The 90-day ED visit rates were 19.0% and 13.6% following inpatient and outpatient hip arthroplasty, respectively [crude RR 0.72 (0.68–0.76); matched RR 0.78 (0.73–0.83)]. Following knee arthroplasty, the ED visit rates were 20.6% and 17.6% following inpatient and outpatient surgeries, respectively [crude RR 0.85 (0.82–0.89); matched RR 0.92 (0.88–0.96)].

**Infection, revision and death.** The 1-year risk of infection, revision, or death was lower in the outpatient setting than the inpatient setting following hip arthroplasty [0.7% versus 1.5%; crude RR 0.45 (0.32–0.63)]. After matching, this association was attenuated but persisted [RR 0.65 (0.45–0.93), p = 0.02]. Following knee replacement, the risk of an event within 1 year was 0.7% for outpatients and 1.0% among inpatients [crude RR 0.74 (0.55–0.99), p = 0.04]. However, after matching, the difference by setting was abrogated [RR 0.90 (0.64–1.26), p = 0.54].

## Discussion

We observed a significant reduction in the volume of hip and knee arthroplasties performed since the start of the pandemic, persisting until at least early 2022. Alongside this reduction in

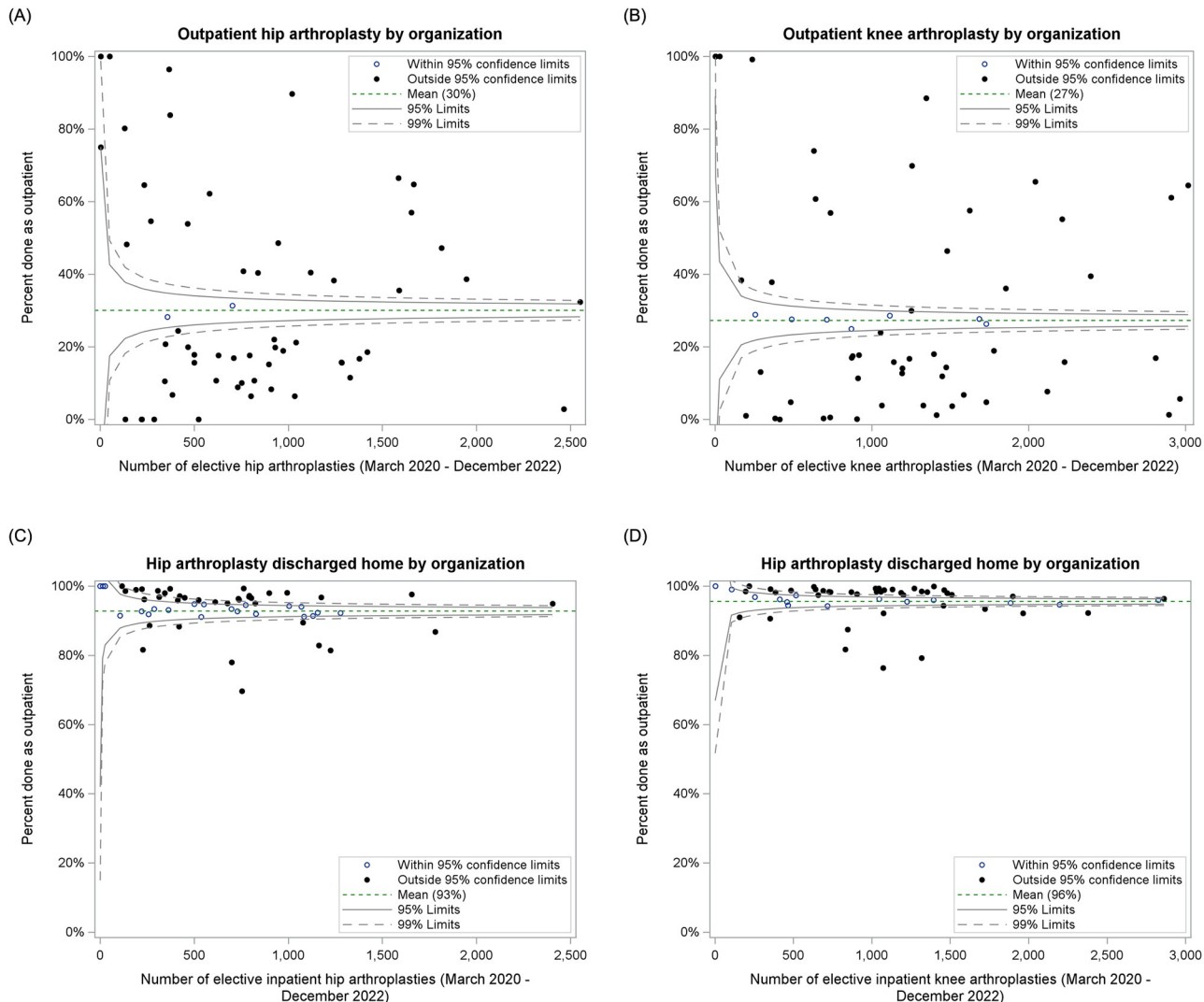

**Fig 3. Funnel plot showing regional variability by hospital where arthroplasty was performed.** (A-B) Percentage of elective hip and knee arthroplasties performed in an outpatient setting during the COVID-19 pandemic (March 2020 to December 2022); (C-D) Percentage of elective hip and knee arthroplasties performed in an inpatient setting who were discharged to home.

inpatient volumes, there was a significant increase during the same period in the volume of outpatient hip and knee arthroplasties, suggesting that hospitals and surgeons responded to system pressures on inpatient bed capacity by shifting arthroplasties to the outpatient setting. While this rapid shift to outpatient care might raise potential concerns around impacts on quality and outcomes, we found in our matched analysis that outcomes were similar or better for outpatient arthroplasties than inpatient cases.

With approximately 50,000 hip and knee replacements performed in Ontario each year (2019 counts), even small improvements in patient outcomes and health system financial outcomes will have a substantial impact on the healthcare system. Other studies have demonstrated similar outcomes with outpatient hip and knee arthroplasty compared with the inpatient setting, including similar complication rates, readmission rates, and patient-reported outcome measures [19–26]. Outpatient surgery can reduce health system resource

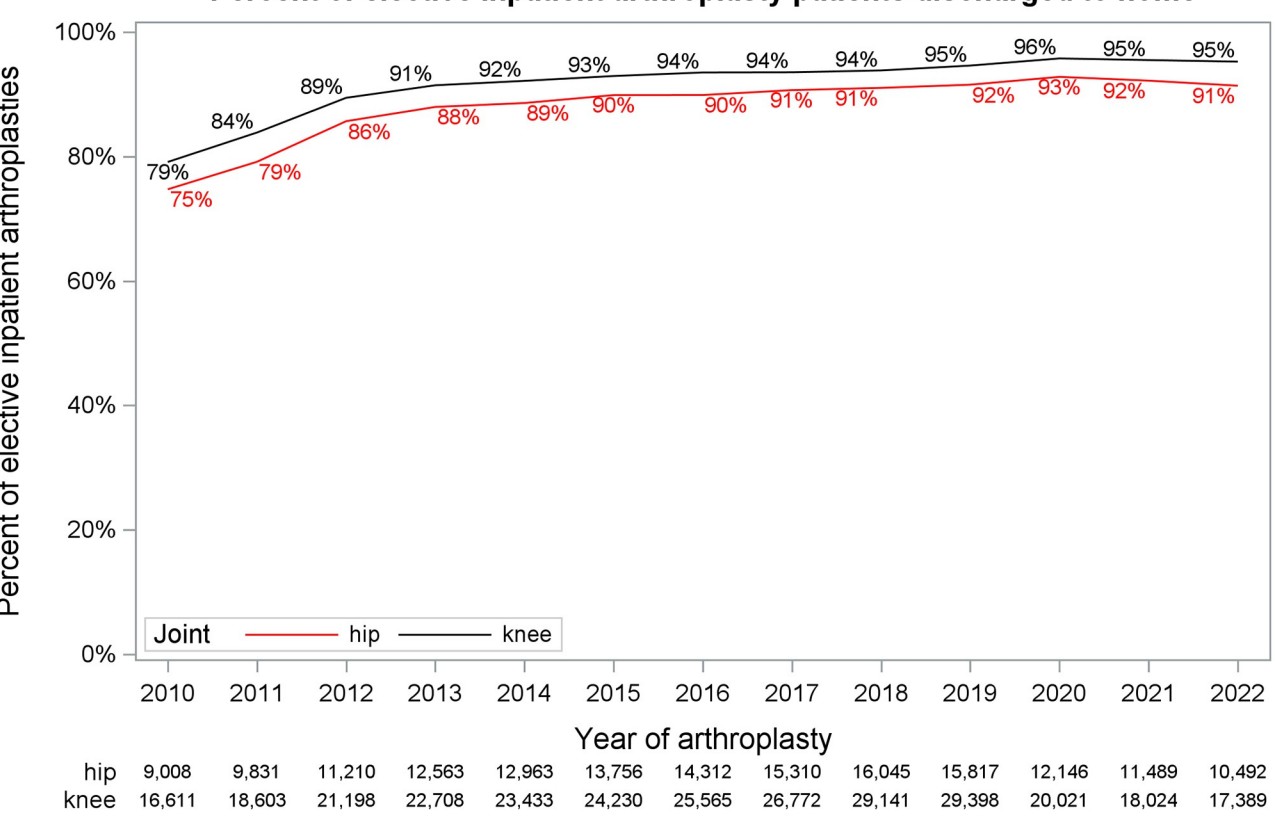

**Fig 4. Inpatient arthroplasties discharge to home.** Percent of inpatient hip and knee arthroplasties discharged to home by year of arthroplasty.

requirements for these cases and free up inpatient beds required for more complex patients. The COVID-19 pandemic triggered the rapid uptake of outpatient arthroplasty in multiple jurisdictions; owing to the positive outcomes observed following this shift, it is likely that this increase in outpatient surgery will be sustained [24, 27–29]. With aging populations, there is expectation that the number of hip and knee arthroplasty procedures will increase, and jurisdictions that still perform inpatient-only procedures may be forced to consider the outpatient setting [30–33].

With evidence mounting that outpatient joint replacement is safe, effective, and cost-saving, efforts should be undertaken to understand the barriers to further uptake. These barriers may explain some of the regional variability observed in this study. One example is patient expectation and hospital culture. Some hospitals have incorporated information about outpatient procedures in educational packages provided to patients, which is important for patient-informed decision-making [34]. Anecdotally, most patients would prefer to recover at home, but facilities must be equipped and organized to provide this service. One consideration is the human resources that are required to operate an outpatient clinic, such as physiotherapist and nurse staffing. Another factor is pain management: some hospitals may be better equipped to provide same-day analgesic medications to support an outpatient model than others. Finally, there are also surgeon-specific factors: anecdotally, younger surgeons may be more likely to perform outpatient procedures.

In an effort to tackle the surgical backlog precipitated by the COVID-19 pandemic, there is mounting interest in the role of ambulatory surgical centres (in Ontario known as

**Table 2. Outcomes by setting during the COVID-19 era for primary elective joint replacements.**

| Hip arthroplasty | Inpatient[a] | Outpatient[a] | Unmatched crude | | Matched crude[e] | |
|---|---|---|---|---|---|---|
| | | | RR (95% CI) | p-value | RR (95% CI) | p-value |
| 90-day readmissions[a,b] | 1,157/23,497 (4.9%) | 247/10,477 (2.4%) | 0.48 (0.42–0.55) | < .0001 | 0.65 (0.56–0.76) | < .0001 |
| 90-day emergency visit[a,b] | 4,462/23,497 (19.0%) | 1,424/10,477 (13.6%) | 0.72 (0.68–0.76) | < .0001 | 0.78 (0.73–0.83) | < .0001 |
| 1-year events[c] | | | | | | |
| Any event[d] | 249/16,765 (1.5%) | 41/6,133 (0.7%) | 0.45 (0.32–0.63) | < .0001 | 0.65 (0.45–0.93) | 0.02 |
| Mortality | 195/16,765 (1.2%) | 29/6,133 (0.5%) | 0.41 (0.28–0.60) | < .0001 | 0.62 (0.40–0.96) | 0.03 |
| Infection or revision | 66/16,765 (0.4%) | 15/6,133 (0.2%) | 0.62 (0.35–1.09) | 0.10 | 0.67 (0.36–1.23) | 0.20 |
| Knee arthroplasty | Inpatient | Outpatient | Unmatched crude | | Matched crude[e] | |
| | | | RR (95% CI) | p-value | RR (95% CI) | p-value |
| 90-day readmissions[a,b] | 1,476/36,394 (4.1%) | 387/13,462 (2.9%) | 0.71 (0.63–0.79) | < .0001 | 0.86 (0.76–0.97) | 0.01 |
| 90-day emergency visit[a,b] | 7,507/36,394 (20.6%) | 2,374/13,462 (17.6%) | 0.85 (0.82–0.89) | < .0001 | 0.92 (0.88–0.96) | 0.0005 |
| 1-year events[c] | | | | | | |
| Any event[d] | 247/25,360 (1.0%) | 52/7,243 (0.7%) | 0.74 (0.55–0.99) | 0.04 | 0.90 (0.64–1.26) | 0.54 |
| Mortality | 206/25,360 (0.8%) | 43/7,243 (0.6%) | 0.73 (0.53–1.01) | 0.06 | 0.93 (0.64–1.36) | 0.71 |
| Infection or revision | 52/25,360 (0.2%) | 16/7,243 (0.2%) | 1.08 (0.62–1.89) | 0.79 | 0.94 (0.51–1.74) | 0.85 |

[a] Outcome was observed within 90 days or 1 year of discharge (inpatient) or registration date (outpatient) following primary arthroplasty.

[b] primary arthroplasties occurred between March 2020 and September 2022

[c] primary arthroplasties occurred between March 2020 and December 2021 (for 1-year follow-up)

[d] composite outcome of infection, revision, or death within 1 year

[e] RR (risk ratio) and 95% CI (confidence interval) using unadjusted modified Poisson regression following propensity-score matching comparing outpatient versus inpatient. Adjusted models are shown for comparison in Appendix 7 for readmissions and emergency department visits, but not 1-year events due to low counts.

"Independent Health Facilities") for performing low-risk operations (e.g. cataracts). Hospitals performing outpatient arthroplasties typically have an inpatient bed on reserve in case the patient is not discharged (e.g. due to pain). Whether joint replacement joins the list of candidate procedures to be conducted in an independent health facility remains to be seen, but results from two systematic reviews demonstrated that most patients (88.1–94.7%) were discharged on the same day, as planned [35, 36]. Despite this, risk prediction models predicting the likelihood of outpatient versus inpatient arthroplasty may have worsened since the pandemic, so independent health facilities must consider the possibility of some inpatient capacity or proximity to an inpatient facility for patients who end up requiring an admission [37].

## Limitations

One limitation is the lack of data on all factors known to select patients for outpatient repair (e.g. frailty, availability of support systems, physical living environment including stairs). Despite this, we matched on comorbidity and various sociodemographic characteristics after restricting to primary elective arthroplasties, which may also incidentally balance the groups on unmeasured confounders of indication. Thus, our findings of equal (or better) outpatient outcomes may be generalizable only to the subset of patients considered appropriate for outpatient arthroplasty [8]. Another limitation is the lack of data on functional outcomes (e.g. quality of life), an understudied topic with mixed results [38]. A further limitation is that our administrative data does not allow us to differentiate inpatient stays on an "intention-to-treat" basis between patients who were originally planned for an outpatient arthroplasty (and were forced to stay overnight due to complications that prevented same day discharge) from patients that were originally planned for an inpatient stay. This limitation may introduce bias

in our comparison of outcomes between the inpatient and outpatient cohorts as outpatient cases that experience poorer in-hospital outcomes are more likely to be shifted to the inpatient cohort.

## Conclusion

In conclusion, the reduction in hip and knee arthroplasty volumes observed in Ontario since the COVID-19 pandemic began was also associated with a dramatic increase in the uptake in outpatient arthroplasty. Ninety-day readmission, 90-day ED visit, and 1-year mortality were similar or better for selected outpatient procedures than for matched inpatient controls.

## Supporting information

**S1 File.**
(DOCX)

## Author Contributions

**Conceptualization:** James Waddell, Erik Hellsten.

**Data curation:** Steven Habbous.

**Formal analysis:** Steven Habbous.

**Investigation:** Steven Habbous.

**Methodology:** Steven Habbous, James Waddell, Erik Hellsten.

**Supervision:** Erik Hellsten.

**Validation:** Steven Habbous.

**Visualization:** Steven Habbous.

**Writing – original draft:** Steven Habbous.

**Writing – review & editing:** Steven Habbous, James Waddell, Erik Hellsten.

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
