## [Decision Letter · Decision Letter 0]

2 Oct 2023

PONE-D-23-22900The Impact of the COVID-19 Pandemic on Outpatient Hip and Knee Arthroplasty in Ontario, CanadaPLOS ONE

Dear Dr. Habbous,

Thank you for submitting your manuscript to PLOS ONE. After careful consideration, we feel that it has merit but does not fully meet PLOS ONE’s publication criteria as it currently stands. Therefore, we invite you to submit a revised version of the manuscript that addresses the points raised during the review process.

We look forward to receiving your revised manuscript.

Kind regards,

Kuo-Cherh Huang

Academic Editor

PLOS ONE

Additional Editor Comments:

Dear Dr. Habbous,

We appreciate your submission to PLoS ONE. Both referees had provided a variety of comments on your manuscript. Here, I would like to bring up a couple of points. Firstly, please consider the nice suggestion from Reviewer 1 in regard to the manuscript title. Secondly, both reviewers had made inquiries and provided suggestions concerning the Methods part. Please respond to them thoughtfully, and explain where you feel you cannot completely agree with reviewers’ suggestions. Finally, I observed that Reviewer 2 had suggested an explicit reference for you to cite. You may wish to consider if the suggested citation is appropriate to be included if the paper fits within your current work. I would like to emphasize that the odds of the acceptance of your submission will not depend on your decision in respect of the suggested citation, however. Thank you.

Academic Editor

Kuo-Cherh Huang

Reviewers' comments:

Reviewer's Responses to Questions

**Comments to the Author**

1. Is the manuscript technically sound, and do the data support the conclusions?

Reviewer #1: Yes

Reviewer #2: Yes

2. Has the statistical analysis been performed appropriately and rigorously? 

Reviewer #1: Yes

Reviewer #2: Yes

3. Have the authors made all data underlying the findings in their manuscript fully available?

Reviewer #1: Yes

Reviewer #2: Yes

4. Is the manuscript presented in an intelligible fashion and written in standard English?

Reviewer #1: Yes

Reviewer #2: Yes

5. Review Comments to the Author

Reviewer #1: I enjoyed the paper and I think it makes a strong argument for the progression to same day surgery.

But I would like to make some suggestions.

1. TITLE - The Impact of the COVID-19 Pandemic on Outpatient Hip and Knee Arthroplasty in Ontario, Canada

I wonder if the title should be more of a "headline" rather than what it is now? You've shown that the pandemic was a huge catalyst for change. And that much of the province HAD to pivot to same day surgery; and that the change was successful with no increase in complication or readmissions.

"The Covid-19 Pandemic: a catalyst to same day arthroplasty across a Province, with no increase in complications or readmissions" This title is just a suggestion, but maybe it grabs me more and makes me feel that I'd want to read it.

or another example..

"The successful and safe in conversion of joint arthroplasty to same day surgery: a Necessity after the covid pandemic".

or maybe even highlight the fact that this conversion occurred in a universal healthcare system as well?

2. Methodology

I am aware of the healthcare across Canada and in particular the Universal Healthcare system. In your methodology, I'm not sure you express how robust the universality is in capturing data. I have read other articles that show this more clearly - and this is the strength of your paper. The fact that you have a single insurer and virtually all hospital and clinic visits are captured by ICES, the DAD and CCI.

This means that your data is capturing virtually all of the half million cases and then virtually all the follow up visits. This is such a unique system and dataset that is not really available elsewhere. I would sing this from the rooftop if it was available to me.

3. Is this the largest series looking at the conversion of arthroplasty to same day surgery? If it is, then please highlight this too.

4. Can you break down where these conversions happened most? i.e. was there a trend for it to occur in just teaching/ academic hospitals or was it even in community hospitals also?

Overall, I found this paper very exciting and the message is that even in a universal healthcare system, hospitals had to shift their practices in order to keep wait times low. This will then show other nations/ healthcare providers there are potentials for the same.

This is where the importance of this paper is.

Reviewer #2: This research analyzes how the provision of outpatient hip and knee arthroplasty in Ontario has changed over time, with a focus on the COVID-19 pandemic period.

The abstract is structured.

The introduction is short and does not achieve a sufficiently good transposition in the topic of the problem. It would be recommended to refer to similar studies from other countries to emphasize the importance of the topic.

As a consequence, the hypotheses of the study should be formulated.

The formulated objective does not detail the analysis period.

The research methodology should briefly describe at the beginning the steps followed and then detail them.

The significance of the volume of data analyzed should be studied.

The discussions are conducted only in relation to the results of this study and do not refer to the results reported in other countries. A comparative recommendation is the recent study:

Moldovan F, Moldovan L, Bataga T. A Comprehensive Research on the Prevalence and Evolution Trend of Orthopedic Surgeries in Romania. Healthcare. 2023; 11(13):1866. DOI: 10.3390/healthcare11131866

But also other similar studies.

The conclusions should indicate the percentage of the monitored variables.

6. PLOS authors have the option to publish the peer review history of their article (what does this mean?). If published, this will include your full peer review and any attached files.

Reviewer #1: No

Reviewer #2: No

---

## [Author Response · Author response to Decision Letter 0]

24 Oct 2023

October 18, 2023

Re: PONE-D-23-22900: The Impact of the COVID-19 Pandemic on Outpatient Hip and Knee Arthroplasty in Ontario, Canada

Dear Dr Huang, 

On behalf of the co-authors, we thank you and the reviewers for the review and suggestions to improve the manuscript. We provide point-by-point responses below. No changes were made to our conflicts of interest or the financial discloser, and for convenience, the original cover letter is included below. Regarding access to data, we are not permitted to share data publicly for legal and privacy reasons, so I believe our data availability statement is appropriate.

Thank you for your time and consideration 

Sincerely,

Steven Habbous, PhD 

Epidemiologist 

Ontario Health (Strategic Analytics) 

Email: steven.habbous@ontariohealth.on.ca

2-Oct-2023

Re: PONE-D-23-22900: The Impact of the COVID-19 Pandemic on Outpatient Hip and Knee Arthroplasty in Ontario, Canada

Dear Dr Habbous,

Thank you for submitting your manuscript to PLOS ONE. After careful consideration, we feel that it has merit but does not fully meet PLOS ONE’s publication criteria as it currently stands. Therefore, we invite you to submit a revised version of the manuscript that addresses the points raised during the review process.

We look forward to receiving your revised manuscript.

Kind regards,

Kuo-Cherh Huang

Academic Editor

PLOS ONE

July 20, 2023

Re: Outpatient hip & knee arthroplasty

Dear editor, 

On behalf of our co-authors, we would like to submit our original article entitled “The Impact of the COVID-19 Pandemic on Outpatient Hip and Knee Arthroplasty in Ontario, Canada” for consideration in your journal. 

Data availability: Ontario Health is prohibited from making the data used in this research publicly accessible if it includes potentially identifiable personal health information and/or personal information as defined in Ontario law, specifically the Personal Health Information Protection Act (PHIPA) and the Freedom of Information and Protection of Privacy Act (FIPPA). Due to these legal and ethical restrictions, data will not be made publicly available. However, upon request, data de-identified to a level suitable for public release may be provided.

Billing: Steven Habbous (lead and corresponding author) is affiliated with Western University, Ontario, Canada. Given the relationship between Western University and PLoS journals, the university can be billed directly if accepted for publication. 

Brief overview: The COVID-19 pandemic has resulted in a marked reduction in elective hip and knee arthroplasty. However, this has also resulted in a shift from a predominantly inpatient procedure (1% pre-pandemic) to 39% (hip) and 36% (knee) by 2022. We demonstrate significant regional variability in the uptake of outpatient arthroplasty. After propensity-score matching, the rate of 90-day readmissions, 90-day emergency department visits, and 1-year composite of mortality, infection, or revision was not different or better following outpatient compared with inpatient arthroplasty. Outpatient hip and knee replacement has the potential to remediate the surgical backlog caused by the pandemic and is likely here to stay.

The contents of this manuscript are not currently under consideration for publication elsewhere. All authors of this research paper participated in the preparation of this manuscript and any conflicts of interest are detailed in the manuscript. On behalf of the authors, thank you in advance for your time and consideration. 

Sincerely,

Steven Habbous, PhD 

Epidemiologist 

Ontario Health (Strategic Analytics)

Email: steven.habbous@ontariohealth.on.ca

EDITOR COMMENTS:

Additional Editor Comments:

Dear Dr. Habbous,

We appreciate your submission to PLoS ONE. Both referees had provided a variety of comments on your manuscript. Here, I would like to bring up a couple of points. Firstly, please consider the nice suggestion from Reviewer 1 in regard to the manuscript title. Secondly, both reviewers had made inquiries and provided suggestions concerning the Methods part. Please respond to them thoughtfully, and explain where you feel you cannot completely agree with reviewers’ suggestions. Finally, I observed that Reviewer 2 had suggested an explicit reference for you to cite. You may wish to consider if the suggested citation is appropriate to be included if the paper fits within your current work. I would like to emphasize that the odds of the acceptance of your submission will not depend on your decision in respect of the suggested citation, however. Thank you.

Academic Editor

Kuo-Cherh Huang

Comments to the Author

REVIEWER COMMENTS:

Reviewer: 1

I enjoyed the paper and I think it makes a strong argument for the progression to same day surgery. But I would like to make some suggestions.

Comment #1: TITLE - The Impact of the COVID-19 Pandemic on Outpatient Hip and Knee Arthroplasty in Ontario, Canada

I wonder if the title should be more of a "headline" rather than what it is now? You've shown that the pandemic was a huge catalyst for change. And that much of the province HAD to pivot to same day surgery; and that the change was successful with no increase in complication or readmissions.

"The Covid-19 Pandemic: a catalyst to same day arthroplasty across a Province, with no increase in complications or readmissions" This title is just a suggestion, but maybe it grabs me more and makes me feel that I'd want to read it.

or another example..

"The successful and safe in conversion of joint arthroplasty to same day surgery: a Necessity after the covid pandemic".

or maybe even highlight the fact that this conversion occurred in a universal healthcare system as well?

Response: These are great suggestions. The transition to outpatient arthroplasty was certainly encouraged by the pandemic. We changed the title to “The successful and safe conversion of joint arthroplasty to same-day surgery: a necessity after the COVID-19 pandemic”

Comment #2: Methodology

I am aware of the healthcare across Canada and in particular the Universal Healthcare system. In your methodology, I'm not sure you express how robust the universality is in capturing data. I have read other articles that show this more clearly - and this is the strength of your paper. The fact that you have a single insurer and virtually all hospital and clinic visits are captured by ICES, the DAD and CCI.

This means that your data is capturing virtually all of the half million cases and then virtually all the follow up visits. This is such a unique system and dataset that is not really available elsewhere. I would sing this from the rooftop if it was available to me.

Response: Thank you, we are certainly privileged to have access to such comprehensive population-based data. We added the following statements to the methods to describe our data sources more fulsomely: 

“All hospitals in Ontario are mandated to report all admission records (discharge abstracts) to the CIHI-DAD. Coding is performed by trained nosologists and routine data quality checks are performed (reference: https://www.cihi.ca/sites/default/files/document/dad-data-quality-current-year-information-2021-2022-report-en.pdf). All surgical outpatient hospital visits are reported to either the CIHI-DAD or CIHI-NACRS as a Level 3 submission, which mandates ICD-10-CA and CCI coding (https://www.cihi.ca/sites/default/files/document/nacrs-data-quality-current-year-information-2021-2022-report-en.pdf).”

Comment #3: Is this the largest series looking at the conversion of arthroplasty to same day surgery? If it is, then please highlight this too.

Response: Following a deeper dive of the existing literature, ours is the largest study to date comparing inpatient with outpatient hip/knee arthroplasty at the population-level from a universal health system. The only other population-based study arose from the United States, which operates under a different health care system. We have therefore added the following to the introduction:

“A systematic review of the literature comparing inpatient with outpatient hip and knee arthroplasty captured studies predominantly from the United States, and studies from other countries were sparse (two from Canada and three from Europe) with small sample sizes (<600).” 

And the following statement as well:

“This is the largest population-based analyses to date from a universal health care system on this topic.”

Comment #4: Can you break down where these conversions happened most? i.e. was there a trend for it to occur in just teaching/ academic hospitals or was it even in community hospitals also?

Response: Interestingly, there was no trend one way or the other for what type of hospitals tended to perform more outpatient procedures. Some high-volume hospitals (e.g. typically academic) performed relative few outpatient procedures, while other performed more. Similarly for and community hospitals and small-volume hospitals, there was no obvious trend. Coloring the points on the funnel plots support this.

Comment #5: Overall, I found this paper very exciting and the message is that even in a universal healthcare system, hospitals had to shift their practices in order to keep wait times low. This will then show other nations/ healthcare providers there are potentials for the same.

This is where the importance of this paper is.

Response: Thank you for your review!

Reviewer: 2 

This research analyzes how the provision of outpatient hip and knee arthroplasty in Ontario has changed over time, with a focus on the COVID-19 pandemic period.

The abstract is structured.

Comment #1: The introduction is short and does not achieve a sufficiently good transposition in the topic of the problem. It would be recommended to refer to similar studies from other countries to emphasize the importance of the topic. As a consequence, the hypotheses of the study should be formulated.

Response: Thank you for raising this point, we agree. After a more thorough review of the literature, we have added the following statement to the introduction to highlight the dearth of evidence on this topic from a population-level:

“…the Centers for Medicare and Medicaid Services in the United States removed from the inpatient-only list total knee replacement in 2018 and total hip replacement in 2020, resulting in a drastic shift towards outpatient arthroplasty.(PMID 36535441, 36898484) Emergency department visits and readmissions were favourable following outpatient hip but worse following outpatient knee replacement. A systematic review of the literature comparing inpatient with outpatient hip and knee arthroplasty captured studies predominantly from the United States, and studies from other countries were sparse (two from Canada and three from Europe) and not population-based (study size <600)…This is the largest population-based analyses to date from a universal health care system on this topic.”

Comment #2: The formulated objective does not detail the analysis period.

Response: The final sentence of the introduction that sites the objective now indicates the study period. As this was an exploratory study, a time period of interest was not pre-specified. Rather, we included historical data as far back as our data permits (2010) while enabling adjustment for covariates like comorbidity that have a 3-year look-back period. We add this reason to the methods.

Comment #3: The research methodology should briefly describe at the beginning the steps followed and then detail them.

Response: We added the following preamble at the beginning of the methods:

“In this retrospective population-based cohort study, inpatient and outpatient hip and knee arthroplasty surgeries were captured between 2010 and 2022. We examined 1) the effect of the pandemic on surgical volumes; 2) trends of inpatient versus outpatient arthroplasty; and 3) outcomes comparing inpatient with outpatient arthroplasty.”

Comment #4: The significance of the volume of data analyzed should be studied.

Response: Following a deeper dive of the existing literature, ours is the largest study to date comparing inpatient with outpatient hip/knee arthroplasty at the population-level from a universal health system. The only other country with similar studies are from the United States, which operates under a different health care system. We have therefore added the following to the introduction:

“A systematic review of the literature comparing inpatient with outpatient hip and knee arthroplasty captured studies predominantly from the United States, and studies from other countries were sparse (two from Canada and three from Europe) with small sample sizes (<600)… This is the largest population-based analyses to date from a universal health care system on this topic.”

Comment #5: The discussions are conducted only in relation to the results of this study and do not refer to the results reported in other countries. A comparative recommendation is the recent study:

Moldovan F, Moldovan L, Bataga T. A Comprehensive Research on the Prevalence and Evolution Trend of Orthopedic Surgeries in Romania. Healthcare. 2023; 11(13):1866. DOI: 10.3390/healthcare11131866

But also other similar studies.

Response: Thank you for sharing this reference. That specific study from Romania forecasts an increasing number of hip/knee arthroplasties over the next decade, so perhaps the Romanian healthcare system can consider outpatient joint replacement for capacity planning (appears to be all inpatient). In this revision, we update the discussion with more recent evidence from the literature with no restriction on country of origin. A systematic review culling the literature until August 2021 identified studies predominantly from the United States (reference #7). The only studies from other countries were small: (n=86 and n=274 from Canada; n=574 from France; n=607 from Netherlands; n=455 from Denmark). A fresh look at the literature did not identify more recent studies except from the United States, which we now cite in the introduction. We added the following to the discussion, citing forecasting studies from the United States, Romania (your suggested article), Taiwan, and Australia:

“With aging populations, there is expectation that the number of hip and knee arthroplasty procedures will increase, and jurisdictions that still perform inpatient-only procedures may be forced to consider the outpatient setting.(PMID 30180053, 37444700, 26318767, 30797228)”

Comment #6: The conclusions should indicate the percentage of the monitored variables.

Response: We have added a “missing” row for Table 1 for variables with missing data. Fortunately the proportion of records with missing data were small (<2%) and non-differential by inpatient versus outpatient setting.

---

## [Decision Letter · Decision Letter 1]

9 Nov 2023

The successful and safe conversion of joint arthroplasty to same-day surgery: a necessity after the COVID-19 pandemic

PONE-D-23-22900R1

Dear Dr. Habbous,

We’re pleased to inform you that your manuscript has been judged scientifically suitable for publication and will be formally accepted for publication once it meets all outstanding technical requirements.

Kind regards,

Kuo-Cherh Huang

Academic Editor

PLOS ONE

Additional Editor Comments (optional):

Reviewers' comments:

Reviewer's Responses to Questions

**Comments to the Author**

1. If the authors have adequately addressed your comments raised in a previous round of review and you feel that this manuscript is now acceptable for publication, you may indicate that here to bypass the “Comments to the Author” section, enter your conflict of interest statement in the “Confidential to Editor” section, and submit your "Accept" recommendation.

Reviewer #1: All comments have been addressed

Reviewer #2: All comments have been addressed

2. Is the manuscript technically sound, and do the data support the conclusions?

Reviewer #1: Yes

Reviewer #2: Yes

3. Has the statistical analysis been performed appropriately and rigorously? 

Reviewer #1: Yes

Reviewer #2: Yes

4. Have the authors made all data underlying the findings in their manuscript fully available?

Reviewer #1: Yes

Reviewer #2: Yes

5. Is the manuscript presented in an intelligible fashion and written in standard English?

Reviewer #1: Yes

Reviewer #2: Yes

6. Review Comments to the Author

Reviewer #1: Thanks for your point-by-point response. I am entirely satisfied with your responses. I think the new changes make this a very readable article with an important message

Reviewer #2: I congratulate you for the work done and the way you followed the suggestions for improving the paper.

7. PLOS authors have the option to publish the peer review history of their article (what does this mean?). If published, this will include your full peer review and any attached files.

Reviewer #1: **Yes: **Amit Atrey

Reviewer #2: No

---

## [Editor Report · Acceptance letter]

14 Nov 2023

PONE-D-23-22900R1 

The successful and safe conversion of joint arthroplasty to same-day surgery: a necessity after the COVID-19 pandemic 

Dear Dr. Habbous:

I'm pleased to inform you that your manuscript has been deemed suitable for publication in PLOS ONE. Congratulations! Your manuscript is now with our production department. 

Kind regards, 

on behalf of

Dr. Kuo-Cherh Huang 

Academic Editor

PLOS ONE